# Moving outside the board room: A proof-of-concept study on the impact of walking while negotiating

Marily Oppezzo[1]*, Margaret A. Neale[2], James J. Gross[3], Judith J. Prochaska[1], Daniel L. Schwartz[4], Rachael C. Aikens[5], Latha Palaniappan[6]

1 Stanford Prevention Research Center, Department of Medicine, Stanford University School of Medicine, Stanford, CA, United States of America, 2 Graduate School of Business, Stanford University, Stanford, CA, United States of America, 3 Department of Psychology, Stanford University, Stanford, CA, United States of America, 4 Graduate School of Education, Stanford University, Stanford, CA, United States of America, 5 Program in Biomedical Informatics, Stanford University, Stanford, CA, United States of America, 6 Primary Care and Population Health, Stanford University Medical Center, Stanford, CA, United States of America

* moppezzo@stanford.edu

## Abstract

### Purpose

Negotiation is a consequential activity that can exacerbate power differentials, especially for women. While traditional contexts can prime stereotypical gender roles and promote conditions that lead to performance differences, these can be mitigated by context shifts. This proof-of-concept study explores whether an easy-to-apply context shift, moving from seated indoors to walking outside, can help improve the quality of negotiated interactions. Here we examine walking's effects on negotiation and relational outcomes as well as experienced emotions, moderated by gender.

### Design

Same-gender pairs were randomly assigned to either sitting or walking as either candidate or recruiter negotiating a job offer.

### Participants

Eighty-one pairs of graduate students or community members participated: sitting pairs: 27 women, 14 men; walking pairs: 23 women, 17 men.

### Intervention

Participants negotiated either while seated (across from each other) or walking (side by side along a path).

### Measures

We measured: negotiation performance (total points) and outcome equity (difference between negotiating party points); subjective outcomes of positive emotions, negative

**Data Availability Statement:** We have published the dataset and the Rmarkdown on figshare at the doi listed: 10.6084/m9.figshare.20645502.

**Funding:** Grant 1K01HL13670201A1 from NIH NHLBI was awarded to MAO. The funders had no role in study design, data collection and analysis, decision to publish, or preparation of the manuscript.

**Competing interests:** The authors have declared that no competing interests exist.

emotions, mutual liking, and mutual trust. With mixed effects models, we tested main effects of condition, gender, and interaction of condition x gender.

## Results

Relative to sitting, walking was associated with: increased outcome equality for women, but decreased for men (B = 3799.1, SE = 1679.9, p = .027); decreased negative emotions, more for women than men (IRR = .83, 95% CI:[.69,1.00], p = .046); and greater mutual liking for both genders (W = 591.5, p-value = 0.027). No significant effects were found for negotiation point totals, positive emotions, or mutual trust.

## Conclusion

This study provides a foundation for investigating easy-to-implement changes that can mitigate stereotyped performance differences in negotiation.

## Purpose

Negotiation is a consequential social activity that takes place in a wide range of contexts, ranging from everyday decision making to formal high-stakes multi-party interactions, each of which can highlight power differentials and prime stereotypically more effective and less effective negotiating behaviors. Physical environments in which the negotiations occur can also influence behavior, with board rooms and stereotypical business items such as briefcases priming competitive behaviors [1]. Situational triggers and role priming increases stereotype threat with more expectations of cooperation from women and competition from men [2].

Research has shown these external influences on negotiation to be modifiable. Altering advocacy, or having a person negotiate on another's behalf vs oneself, and having multiple issues to negotiate vs one issue improves women's negotiation behavior and removes a gender gap in performance [3]. Here we consider whether an easy-to-implement context switch can decrease the stereotype threat and priming of social role expectations associated with a common negotiation task, job offers. In this proof-of-concept study, we look at the potential impact of moving the negotiation, historically researched seated, facing each other, in an office environment, to walking together on an outdoor path, with both parties on equal footing. Specifically, we investigate whether walking while negotiating can make a difference in either the outcome of a negotiation or the participants' emotional and social experience of the interaction, and how these effects might differ by gender.

### Gender and negotiation

Research on gender and negotiation has found that while there is a small effect of gender and gender role expectations, context shifts can reverse these [3]. Performance differences can occur when gender triggers prime gender role expectations, stereotypes, and power differences [3, 4]. For example, women performed worse than men when reading that negotiation success was linked to assertiveness and self-interest advocacy and poorer performance linked to being emotional or accommodating [5]. Further, mere exposure to stereotypical business objects, such as a boardroom and briefcases, primes competition and decreases perceptions of cooperativity in social situations [1]. Women still have lower economic power, and job negotiation contexts prime these hierarchical gaps [6, 7]. One study showed if recruiter and candidate

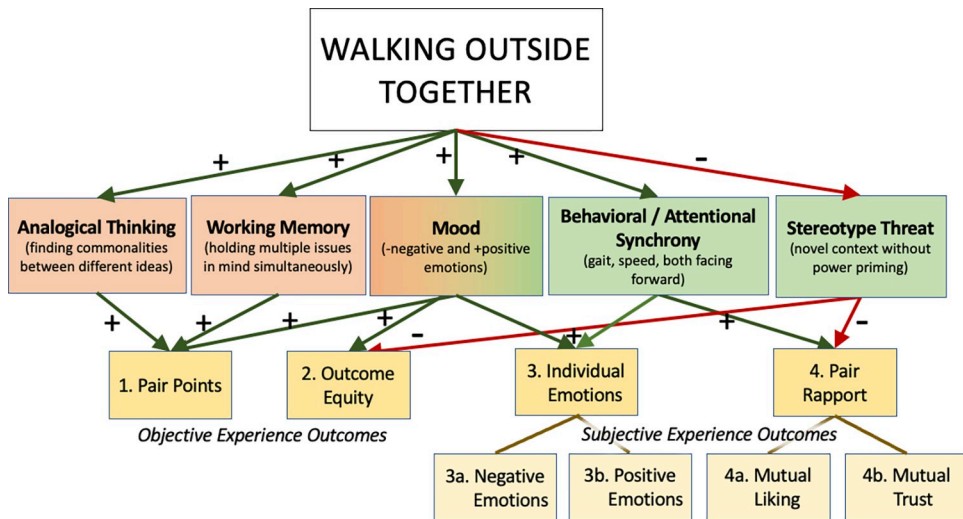

**Fig 1. Hypotheses concept map.** Walking-specific benefits are indicated by the orange color or tint, and walking outside together benefits are indicated by green color or tint. The small to medium effects of gender on negotiation outcomes are ameliorated by contextual shifts, therefore we expect the effect of walking outside together will particularly benefit women.

seated opposite a desk from each other with a difference in chair height, women negotiation pairs showed poorer economic outcomes; however, shifting the arrangement to be more egalitarian with the parties seated side-by-side mitigated this decrement [8]. Similarly, other small contextual shifts, such as when negotiations are shifted to have a woman negotiating on behalf of another [9], or the negotiation is integrative with multiple issues at stake [10], also mitigate performance differences.

Here we explore the impact of an easy-to-implement contextual shift of changing the negotiation physical environment by walking together outside. We describe below how this context shift may influence both objective and subjective negotiation outcomes, and when relevant note how these may particularly help close the contextually-influenced gender negotiation gap [3]. Fig 1 displays a conceptual map of how walking outdoors may improve negotiation outcomes. For ease, in the text below the concepts are **bolded** and the outcome measures for the current study are *italicized*.

## Cognitive negotiation benefits of walking

We chose walking as a physiological change to the negotiation context for several reasons. First, we wanted to investigate whether—independent of gender—walking might improve negotiation. Research has found that walking has many cognitive benefits, including two particularly relevant to negotiation. One is **analogical thinking**; one study found walking improved people's abilities to make analogies [11], which requires identifying a common structure between two different ideas [11]. Optimal negotiation outcomes often require integrating disparate perspectives and finding common ground [12], as opposed to focusing only on a single viewpoint [13]. The second is **working memory**. One study found walking at a self-selected pace improved working memory compared to sitting [14]. Negotiating complex agreements across multiple issues requires keeping those issues in mind simultaneously. We choose a multi-issue negotiation activity with novel solution potential as a measure sensitive to these cognitive benefits of walking (performance measured by *total points* earned in the negotiation).

## Affective negotiation benefits of walking

Walking has been shown to improve affect and **mood**. For example, a study on active workstations found increased psychological arousal, decreased boredom, and lower appraised task stress compared to sitting or standing [15]. Walking outside is also associated with an improvement in mood [16, 17]. Walking's mild intensity and rhythmic quality as an activity may particularly benefit positive affect [18]. In negotiations, displaying positive emotions improves social and economic negotiation payouts [19], while high anxiety can hurt [20]. Additionally, women report disliking negotiation more than men; women prefer to avoid negotiation, and when they do engage in negotiation they report feeling anxiety and discomfort [21–24]. We measure both *negative and positive emotions* after negotiation to detect whether walking while negotiating mitigates negative or enhances positive emotions.

Walking together along a predetermined path increases **behavioral and attentional synchrony**. The common negotiation context is seated face-to-face, and in job negotiations, often with a higher-powered party sitting behind their desk or a table with a lower-powered party seated across from them on the other side of it; shifting to being seated side-by-side decreases power-differential performance detriments [8]. Walking together also has both parties side by side, with joint attention on the path, and increases behavioral synchrony, including speed, gait, and path trajectories [25, 26]. Behavioral synchrony has been linked to both rapport building [27], *mutual liking* [27–29], *mutual trust* [27–29], and cooperation, independent of positive feelings [30]. Even perceived synchrony has been shown to increase empathy [31], and can prime people to feel as if they are cooperating together rather than competing; this is important for sharing perspectives and may improve *outcome equity*, or mutual winning, in negotiation outcomes.

Finally, walking together outside is a novel, non-stereotypical context for a negotiation. As noted earlier in the section on gender and negotiation, primes such as seated face-to-face, or office room cues such as suitcases can prime stereotypical behavior of competition and prime differences in economic power (more for women) [3–6]. Active, non-sedentary work environments are also linked to reduced feelings of territoriality [32], which primes competition and zero-sum behaviors in negotiation. Indoors face-to-face with negotiation can prime the negative experience of negotiation women report disliking [21]. Finally, environments that prime competition lead to less cooperative, value-creation in negotiations, and competitive negotiation behaviors are associated with less *mutual trust* [33]. We expected walking outside on common ground to remove the **stereotype threat** cues towards competition and power, which might increase cooperative behavior to improve both *outcome equity* and pair rapport (measured by *mutual liking* and *mutual trust*).

## The present study

The goal of this study was to investigate whether walking outside (compared to sitting inside) would improve several negotiation-related outcomes, and to explore the potential for differential effects of walking while negotiating by gender. We randomly assigned same-gender pairs to either sitting across from each other indoors or walking side by side outside performing the "New Recruit," a multi-issue job negotiation exercise with a candidate and recruiter. We measured both objective negotiation outcomes, *total points* and *outcome equity*, as well as subjective negotiation outcomes of individual emotions (*negative* and *positive emotions*) and pair rapport (*mutual liking* and *mutual trust*). Our overall hypothesis was that walking would enhance objective and subjective negotiation outcomes, and that this effect would be particularly enhanced for women. Specifically, we hypothesized that compared to negotiating sitting together inside, negotiating when walking together when outside would lead to improvement

in: objective negotiation outcomes of *total points* (H1) and *outcome equity* (H2); subjective individual negotiation outcomes of *negative emotions* (H3a) and *positive emotions* (H3b); and subjective pair outcomes of *mutual liking* (H4a) and *mutual trust* (H4b). For each outcome, we expected effects of walking compared to sitting to be greater for women pairs than men pairs.

## Methods

### Design and hypotheses

In a between-groups design, same-gender pairs were randomly assigned to either sitting or walking conditions, and randomly assigned to a role of either candidate or recruiter in a multiple-issue job negotiation exercise, "New Recruit" [34]. Same gender pairs were employed to maximize power to detect hypothesized gender effects (versus complex effects of mixed gender dyads) [21] We measured both objective and subjective negotiation outcomes. The New Recruit measure has role-specific point payouts for each of eight different issues within the negotiation task. Objective outcomes, related to negotiation performance, include: *total points* which the pair earned together (pair); and the *outcome equity*, or distribution of these points between the individual parties serving as a proxy for the difference in power (pair). Subjective outcomes, both individual and social, include: *negative emotions* and *positive emotions* (individual); and *mutual liking* and *mutual trust* within the pair (pair). For each negotiation outcome, we looked for the main effect of condition (sitting vs walking), the main effect of pair gender (men vs women), and the interaction (condition x gender).

### Ethics statement

The Stanford University Institutional Review Board approved this study, and all participants signed a consent form prior to participation.

### Sample

From a graduate business school participant pool, 286 eligible participants signed up, with 86 cancellations, and 19 pairs excluded due to participants having done the activity before in a class, not complying with study protocol, or knowing their counterpart personally. The latter was a reason for exclusion because we did not a priori assess or randomize participants based on knowing each other, which would influence the relational outcomes of mutual liking and mutual trust. We also wanted to isolate the effects of walking independent of prior relationship (and did not have a pre-test measure to assess change). Final analyses used 162 adults (81 negotiating pairs: 31 men, 50 women).

### Intervention

**Procedure.** Participants were run as pairs, randomly assigned on a 1:1 basis to either the walking or sitting condition, and then role (candidate or recruiter) via online random number generator (random.org). After written informed consent, participants were given the negotiation packets for their role, and shown a video describing the task rules of engagement (e.g., do not share individual point payout values with the other participant). They then were given ten minutes to read and become familiar with their priorities, after which condition-specific instructions were provided.

Each pair was given 30 minutes to complete the negotiation, with clipboards to hold their packets, folders to cover up pay out schedules, and timers, set for both a five-minute warning

and the 30-minute mark at which to stop. If they completed the negotiation early, they were asked to pause the timer and alert the experimenter.

Sitting condition participants sat across from each other, face-to-face in a room, with a table in between them. All sitting conditions were run in the same room. Walking condition participants were given a map of a 15-minute walking loop outside to follow. The loop was on the university campus which is a mix of trees, university buildings, alternate walking paths, and people. Variations in busy-ness, sunshine, and temperature occurred in the outdoor walking condition.

After the negotiation, each participant took the ten-minute post-survey, was compensated $25, and debriefed.

**Negotiation task.** The negotiation activity was the "New Recruit" task, an 8-issue, dyadic, simulated job offer exercise in which one person plays the role of a recruiter and another, the candidate [34]. Parties must agree on eight issues: two "fixed-sum" issues where parties' pay out are mutually opposing (one party's gain comes at the other's loss); two congruent issues, or "value sharing" issues where parties' interests are aligned (both parties want the same thing); and four integrative, or "value-creating" issues where parties can make trade-offs to create value and earn more points through pairing issues with asymmetric payouts. Reaching optimal agreement depends upon information sharing and cooperation. The measure is suited for determining if the cognitive benefits of walking could extend to negotiation in two ways. First, finding a novel way to package multiple issues may capture the creative benefits of walking. Second, maintaining multiple issues and points of view may capture walking's working memory benefits.

The exercise affords various performance measures; we assess the overall performance by the total pair points, and the equality or fairness of outcome by point distribution, or subtracting the individual point totals from each other (e.g., of 10,000 total points how even was the split between recruiter and candidate).

## Measures

**Objective negotiation outcomes.** *Total points (Pair, hypothesis 1).* A pair or individual's points could range from -8400 to 13,200, summing the outcomes for all issues for both parties at the pair level.

*Outcome equity (Pair, hypothesis 2).* The difference between each participant within the pairs score (candidate total subtracted from recruiter total) measured the equality of the final contract and served as a proxy for the power differential of the outcome and between the two parties. Differences closer to zero indicate more equal distribution of the point payout. A very unequal distribution of points, or a large difference, suggests more objective power differential, where one party took more at the other party's expense.

**Subjective negotiation outcomes.** *Negative and positive emotions (Individual, hypotheses 3a and 3b).* To test the emotional impact of walking during negotiation, we measured both positive and negative emotions. Negative (17) and positive (7) emotions were evaluated on a 5-point Likert scale from not at all to extremely, using emotion words from the short form of the Profile of Mood States [35] and a subset of the Positive and Negative Affect Schedule [36]. Negative emotion ratings and positive emotion ratings were averaged for each individual.

*Mutual liking and mutual trust (Pair, hypothesis 4a and 4b).* On a five-point Likert scale from not at all to extremely, participants were asked the following questions: 1. How much did negotiating with your negotiation partner make you feel a sense of liking for them? 2. How much do you trust your negotiation partner? These questions represented liking and trust, respectively. To reflect mutual, pair liking and trust, these individual scores were averaged.

## Data analysis plan

For each negotiation outcome, we tested the main effect of condition (sitting vs walking), the main effect of gender (men vs women), and whether the effect of condition differed by gender. Total points, outcome equity, mutual liking, and mutual trust were analyzed at the pair level. For normally distributed continuous outcome of outcome equity, we used linear regression with condition, gender, and condition x gender as predictors. For the non-normally distributed ordinal outcomes of total points, mutual liking, and mutual trust, we used non-parametric Mann-Whitney U test for condition, and then sub-analyses for the effect of condition for each gender. The Mann-Whitney U is a robust test which does not require normal distribution assumptions and works for ordinal data such as Likert scales used for the mutual liking and trust outcomes.

Positive and negative emotions were analyzed at the individual level using hierarchical linear modeling [37]. A random intercept for pair was included to model nonindependence within the pair. For the continuous, Poisson-distributed negative and positive emotions (individual), we used linear mixed effects models including fixed effects for condition, gender, and condition x gender, and a random intercept for pair, with R's glmer function, specifying a Poisson error distribution and a log link function. All data were analyzed using R [38] using the stats and lme4 [39] packages.

## Results

### Sample demographics

The participant pool is largely drawn from the student population. Of the original pool of 200 adults who participated in the current study, 180 provided demographic data. Race/Ethnicity percentages were as follows: 32.8% (n = 59) Caucasian, 20.6% (n = 37) East Asian American, 12.8% (n = 23) Multiracial, 12.2% (n = 22) Hispanic, 11.1% (n = 20) African American, 7.2% (n = 13) South Asian American, and 3.3% (n = 6) Other. The pool was 91.7% students, with the remaining unspecified either as alumni or other participants from the community. Demographic data was only provided at screening and not linked to outcome.

The mean (SD) of the total points for the entire sample was 9995 (1952.7), and the data for this outcome was non-normally distributed; therefore non-parametric analyses were used.

### Objective negotiation outcomes

**Total points (Hypotheses 1).** Using the Mann-Whitney U test to compare the total points between pairs who walked vs pairs who sat, we found no main effect of condition, W = 769.5, p = .63. The Hodges-Lehmann, a nonparametric estimate for effect size, = -.0000057, 95% CI: [−1200, 600]. The subgroup analyses comparing total points between conditions was not significant for women pairs, W = 285, p = .60. Hodges-Lehmann = -.000026, 95% CI: [−1200, 600]; or for men pairs, W = 117.5, p = .97. Hodges-Lehmann = -.000032, 95% CI: [−1200, 1200]. While walking had no significant effect on total points compared to sitting, the means were in the predicted direction: means and standard errors for sitting pairs: 9795.0 (332.9); walking pairs:10190.2 (280.5).

**Outcome equity (Pair, hypothesis 2).** Multiple regression analysis was used to test whether condition, gender, or condition x gender significantly predicted the outcome equity. There were no main effects of condition (B = -1839.5, SE = 1321.8, p = .17) or gender (B = -2041.8, SE = 978.9, p = .57). The interaction of condition x gender (B = 3799.1, SE = 1679.9, p = .0266) was a significant predictor. Means and SEs are shown in Fig 2. Walking significantly increased the outcome equity between the roles within a pair for women, however walking significantly decreased the difference for men.

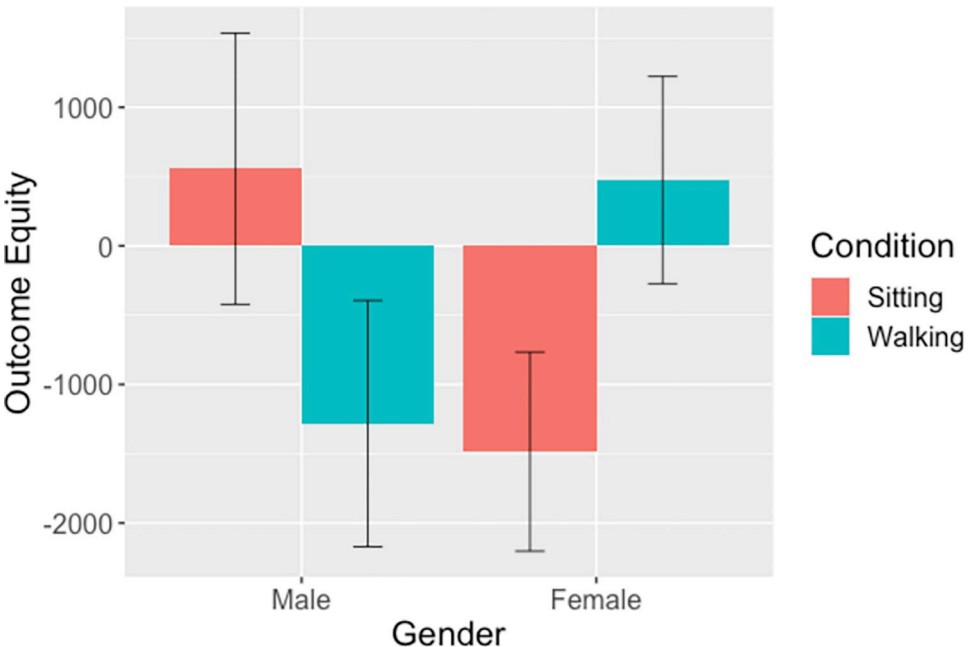

**Fig 2. Outcome Equity x Condition x Gender.** *Note*: Error bars reflect standard errors of the mean. Closer to zero, or lower values of the differences, indicates more equal distribution of points within the pair.

## Subjective negotiation outcomes

**Negative emotions (Hypothesis 3a).** The linear mixed model showed no main effects for either condition (IRR = 1.07, 95% CI: [.93,1.23], p = .36) or gender (IRR = 1.06, 95% CI: [.93,1.22], p = .33), but did show a significant interaction between condition and gender (IRR = .83, 95% CI: [.69, 1.00], p = .0456). For women, negative emotions were lower for those who walked compared to those who sat; for men, negative emotions were higher for men who walked (Fig 3).

**Positive emotions (Hypothesis 3b).** The linear mixed model showed a significant main effect of gender (IRR = .81, 95% CI: [.69-.95], p = .0103), but no significant main effect of condition (IRR = .99, 95% CI: [.83–1.18], p = .937) or significant interaction of condition and gender (B = 1.06, 95% CI: [.84–1.32], p = .618). Regardless of condition, men reported significantly higher positive emotions than did women (Fig 4).

**Mutual liking (Hypothesis 4a).** Using the Mann-Whitney U test to compare the average liking between pairs who walked vs pairs who sat, we found a main effect of condition, W = 591.5, p-value = 0.027. The Hodges-Lehmann, non-parametric estimate of effect size = -.50, 95% CI: [-.50, -.000059]. The interpretation of the Hodges-Lehmann is a difference in the medians of the two groups, indicating the difference in mutual liking between those who walked and those who sat was about ½ point on the Likert scale. The subgroup analyses comparing point performance between conditions for women pairs was not significant, W = 253.5, p-value = 0.2505. Hodges-Lehmann = -0.50, 95% CI: [-.050, -.000031], but for men pairs it was significant, W = 68.5, p-value = .03884. Hodges-Lehmann =.-0.50, 95% CI: [-1.0, -.0000027]. Fig 5A shows means and standard errors of average liking, indicating that for both genders walking increased the average liking.

**Mutual trust (Hypothesis 4b).** Using the Mann-Whitney U test to compare the average trust points between pairs who walked vs pairs who sat, we found a main effect of condition, W = 737, p-value = 0.4249. The subgroup analyses comparing average trust between

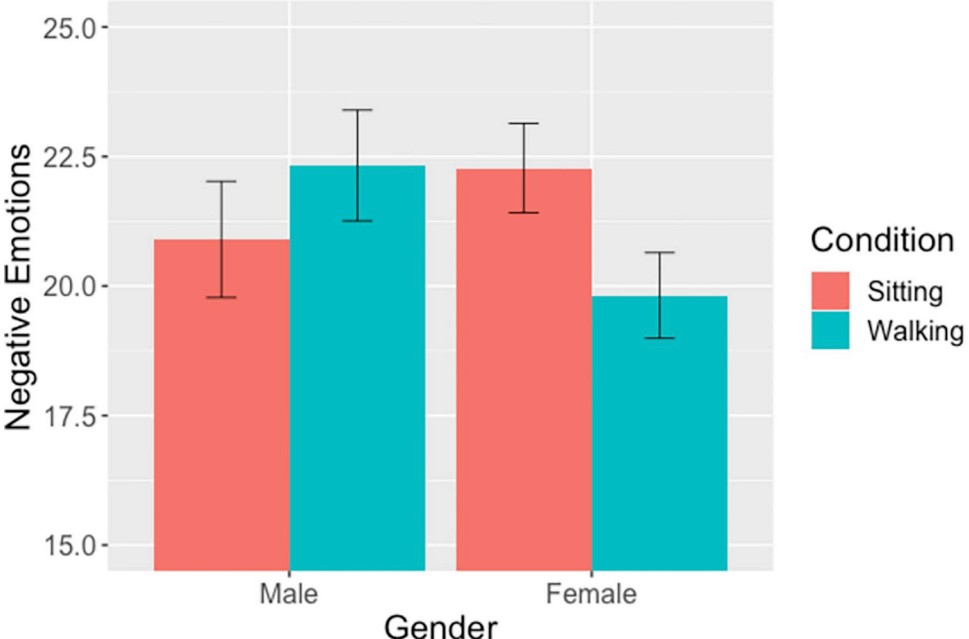

**Fig 3. Negative Emotions x Condition x Gender.** *Note*: The means reflect estimated marginal means. Error bars reflect standard errors of the mean.

conditions for women, W = 271.5, p-value = 0.4273 and for men, W = 114, p-value = .8532 was not significant in either case.

Using the Mann-Whitney U test to compare the average trust points by gender. Here, we found a main effect for gender, W = 999, p = .02616, Hodges-Lehmann = .50, 95%CI:

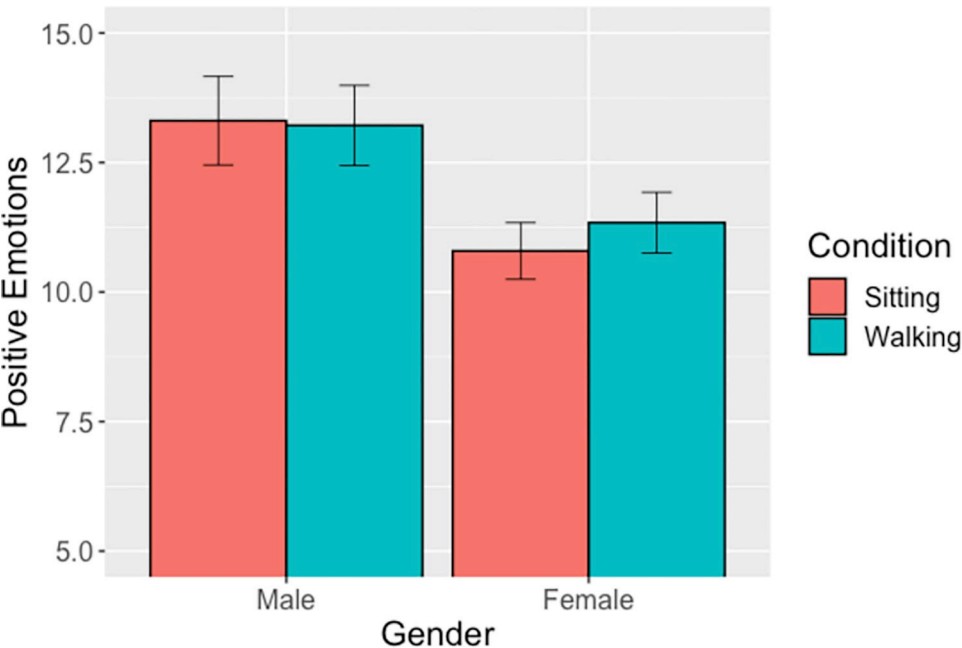

**Fig 4. Positive Emotions x Condition x Gender.** *Note*: The means reflect estimated marginal means. Error bars reflect standard errors of the mean.

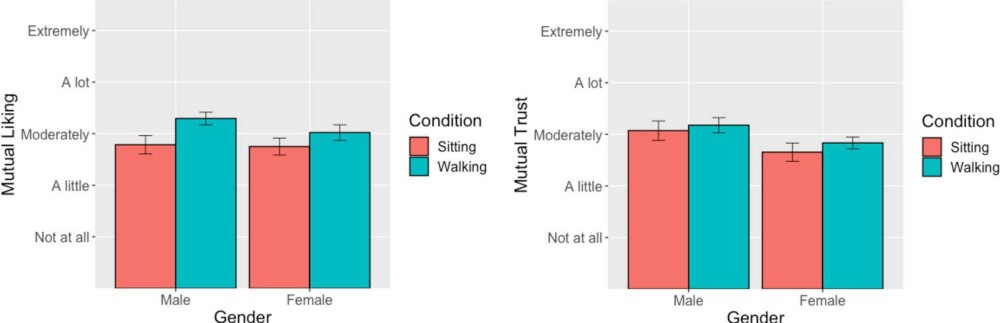

**Fig 5. a.** Mutual Liking x Condition x Gender. *Note*: Error bars reflect standard errors of the mean. **b.** Mutual Trust x Condition x Gender. *Note*: Error bars reflect standard errors of the mean.

[.00007, .5]; this indicates that the men pairs, compared to women pairs, had on average higher mutual trust by ½ a Likert scale point. The subgroup analyses comparing average trust between gender for those sitting, W = 232, p = .1515, Hodges-Lehmann = .50, 95%CI: [-.000043, 1.0], and those walking, W = 263.5, p = .1049, Hodges-Lehmann = .50, 95%CI: -.000021,.05], was not significant. Fig 5B shows that men pairs had higher trust than female pairs, regardless of condition.

## Discussion

Walking together has been suggested to benefit some social processes. (For an excellent theory paper on the linguistic connection between walking together and conflict resolution, (e.g. "moving on", "finding common ground", being "at a stand-still"), see Webb et al., 2017 [40]). As a proof-of-concept we explored the potential of this simple context shift–walking together outside—to improve objective and subjective negotiation outcomes.

We selected a negotiation exercise between candidate and recruiter that had opportunities for both cooperative and competitive behavior. The only gender-specific improvements of walking were improved outcome equity (our proxy for power differential) and decreased negative emotions. Walking was associated with mutual liking independent of gender, while gender (men) was associated with more positive emotion and mutual trust (compared to women) independent of walking. These findings show more of a relational impact of condition than a work performance impact. We discuss the implications and limitations of these results below.

### Implications

Negotiation is commonly associated with more negative feelings for women, but walking outside together led to less negative emotions for women. Additionally, unequal outcomes in negotiations can be perpetuated by stereotype threat of gender-based behaviors and priming of different power positions; walking outside together led to more outcome equity, therefore a more equal sharing of points. One explanation for both of these results could be that walking created a more cooperative milieu, as walking together is a synchronous activity [25, 26], and removed the negative impact of stereotypical environments (e.g. office board rooms). Having a novel environment and less negative emotions could have led to a more equitable more cooperative performance outcome.

Additionally, walking together provided an informal context. This stands in contrast to the competitiveness and power differences that are typically associated with negotiation performance, stereotypical male behavior, and the more formal contexts (e.g. seated in a boardroom,

one person across a desk from the other) where they take place [1, 3, 4]. When traditional gender roles are primed, women show poorer performances [4]; but outside on a partnered walk, there are few, if any, cues to prime social roles or competitive, zero-sum behaviors.

That walking decreased the outcome equity in men, a higher level of power differences in the outcomes, was surprising. While exercise may increase testosterone in men more than women, theoretically priming more competitive behaviors [41, 42], walking is not at the level of intensity as aerobic exercise. One possibility is that walking together for women motivates a cooperative mindset but walking together for men feels more novel, or may create greater vigilance and more competitiveness. Future studies should not only include measures of perceived power and perceived competitiveness, but also include a measure of walking speed, which may be faster in more competitive negotiations.

The psychological impact of walking for women was perhaps the most important result. Women tend to have more negative feelings about negotiating than men [20, 24]. Our current study showed men felt more positive in both negotiation conditions. To the degree that negative emotions compete with cognitive resources needed for successful negotiations, walking together may provide a simple solution for women leveling the psychological playing field. Further, walking increased mutual liking between the parties for both men and women. As many negotiations in real life require future interactions, an outcome with less negative feelings and more mutual liking may be an important one, a future benefit not often considered or relevant in single-instance negotiation studies. Future studies should test multiple meeting negotiations to see if this increased rapport confers sustainable benefits.

Finally, it is worth noting that though walking did not affect the pair's total points, it also did not hurt performance. From a public health perspective, finding ways to interrupt prolonged sitting with thoughtfully chosen walking meetings, without hurting performance, is important to decrease the negative health impacts of prolonged sedentary behavior [43, 44]. Researching different social interactions and meeting types that can be done while walking, without hurting performance, can thoughtfully motivate decisions for integrating activity into sedentary workdays which do not negatively affect worker productivity.

## Limitations and future directions

The study had several limitations. First, this was a proof-of-concept study exploring the potential impact of a novel context variable (i.e. a walking meeting) on outcomes of interest. We did not do an a-priori statistical power analysis and we encourage readers to interpret our results as exploratory research. All findings should be taken as preliminary and offering inspiration for further, focused studies. Statistical power is a serious limitation for studies in this context due to high variances and low likely effect sizes [45]. As a power sensitivity analysis, based on the variances in total points we observed, for example, a study of this size (~ 40 pairs in each condition) would be powered to detect a difference by condition of approximately 1230 points (this is the minimum detectable effect). In particular, our preliminary finding that the impacts of walking on negotiation outcomes may differ by gender represents an interesting possible focal point for follow-up. If the effects of walking are indeed opposite for men and women, then statistical power may be better in single-gender studies than in studies (like this one) which consider both genders and try to estimate an overall effect.

Another limitation is our post-test-only assessment of emotions. Though randomization helps mitigate effects of any individual differences in mood, future research should have a pre-post design for a more rigorous test for the impact of walking on emotions from negotiations. A strength of post-only assessment is that it avoids priming (e.g. extra attention to their emotions) or cuing participants to study hypotheses.

The largely homogeneous graduate school of business student population make the outcomes less generalizeable; students may have brought stronger narratives to either recruiter or candidate roles having more exposure to job negotiation themes in their business classes. A more representative adult sample can test if results generalize to workforce populations or non-financial negotiations.

In contrast to some literature [3, 4] we found no significant difference in the total points by condition for men vs women pair negotiations. One reason for this could be that New Recruit had more cooperative task features, shown to moderate the gender differences [3, 4, 46]. A stronger test of whether walking can mitigate effects of harmful context-cues may be to induce a stereotype threat prior to the negotiation [4, 5, 9, 47, 48].

We did not dissociate the impact of walking from walking together. Future iterations of this work might also include a walking virtual (computer-mediated) meeting to isolate whether walking together in person confers differential benefits to walking in general. To isolate the cognitive effects of walking, a study could contrast single and multiple issues (working memory), or assess the number of divergent viewpoints discussed and resolved during the negotiation process (analogical thinking).

We also did not isolate the effects of being outdoors on the current outcomes. An ideal 2x2 future study could vary outdoors (inside board room vs outside board room) by movement (sitting vs walking). While there is substantial literature supporting the mood enhancing effects of being outdoors [49, 50], many of the cognitive effects of movement relevant to the current negotiation exercise is movement whether one is indoors or outdoors [11, 51]

It is possible that walking together was simply a novelty. Instead of the shared physical movement and path navigation focus driving the effects, the key feature may be being somewhere other than inside an office. Additional conditions of sitting side by side outside on a bench or inside at a coffee shop can dissociate the novel, non-traditional-business context from the walking together. To distinguish between synchronous and non-synchronous physical movement, playing golf is a feasible condition with real-world precedence as a context for work meetings.

The social and psychological findings from this study motivate researching the longer-term impact of walking together. Perhaps it does not improve immediate negotiation performance, but rather leads to better future negotiations and social rapport between the negotiators. Using measures where a second meeting for either the same or a different negotiation with the same partner is a logical next step.

Notably, this study is the first of its kind to test the effect of the simple contextual change of walking together to decrease historical differences in negotiation outcomes for men and women. Compared to sitting, walking while negotiating increased equality of outcomes and decreased negative emotions for women, but not for men. It also led to increased liking for both women and men. If walking together could help mitigate performance differences in negotiation, this is an easy step forward towards a more equal and shared footing in the workplace for men and women. Importantly, finding ways to integrate light movement into work, especially into emotionally charged situations, without decreasing performance can lead to healthier work cultures for all.

## Author Contributions

**Conceptualization:** Marily Oppezzo, Margaret A. Neale, Judith J. Prochaska, Daniel L. Schwartz, Latha Palaniappan.

**Data curation:** Marily Oppezzo, Margaret A. Neale, Rachael C. Aikens.

**Formal analysis:** Marily Oppezzo, Judith J. Prochaska, Daniel L. Schwartz, Rachael C. Aikens.

**Funding acquisition:** Margaret A. Neale, Latha Palaniappan.

**Investigation:** Marily Oppezzo, Margaret A. Neale, Judith J. Prochaska, Daniel L. Schwartz, Latha Palaniappan.

**Methodology:** Marily Oppezzo, Margaret A. Neale, Judith J. Prochaska, Daniel L. Schwartz, Latha Palaniappan.

**Project administration:** Marily Oppezzo, Margaret A. Neale, Judith J. Prochaska, Latha Palaniappan.

**Resources:** Margaret A. Neale, Judith J. Prochaska, Rachael C. Aikens, Latha Palaniappan.

**Software:** Rachael C. Aikens.

**Supervision:** Marily Oppezzo, Margaret A. Neale, James J. Gross, Judith J. Prochaska, Latha Palaniappan.

**Visualization:** James J. Gross, Daniel L. Schwartz.

**Writing – original draft:** Marily Oppezzo, James J. Gross.

**Writing – review & editing:** Marily Oppezzo, Margaret A. Neale, James J. Gross, Judith J. Prochaska, Daniel L. Schwartz, Rachael C. Aikens, Latha Palaniappan.

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
