## [Decision Letter · Decision Letter 0]

11 Oct 2022

PONE-D-22-23834Weaving activity into work: Walking while negotiatingPLOS ONE

Dear Dr. Oppezzo,

Thank you for submitting your manuscript to PLOS ONE. After careful consideration, we feel that it has merit but does not fully meet PLOS ONE’s publication criteria as it currently stands. Therefore, we invite you to submit a revised version of the manuscript that addresses the points raised during the review process.

We look forward to receiving your revised manuscript.

Kind regards,

Ricky Siu Wong, PhD

Academic Editor

PLOS ONE

Journal Requirements:

2.We note that you have stated that you will provide repository information for your data at acceptance. Should your manuscript be accepted for publication, we will hold it until you provide the relevant accession numbers or DOIs necessary to access your data. If you wish to make changes to your Data Availability statement, please describe these changes in your cover letter and we will update your Data Availability statement to reflect the information you provide.

Additional Editor Comments:

Dear Dr. Oppezzo:

I write to you regarding manuscript # PONE-D-22-23834 entitled " Weaving activity into work: Walking while negotiating" which you submitted to Plos ONE.

The paper was reviewed by myself and two reviewers. Thank you for submitting your manuscript to our journal. The paper was reviewed by a very qualified team. They all come to the same conclusion. It looks like the paper would need to undergo a rather substantial revision and rewrite of major sections. While there is certainly no guarantee about a positive outcome of this new submission, we encourage you to consider doing so. We are therefore leaving it up to you of whether you would like to take this route, or rather move to a different journal. Reviewers are clear in their criticism, and I concur with their conclusion in my recommendation to the EIC. I hope you will find the feedback in this review process helpful. Please refer to the following points if you decide to revise the manuscript.

1. Both reviewers and I have identified that a detailed explanation as to why, exactly, walking would be more beneficial for women negotiators is missing. As pointed out by R1, why, specifically, should pairs of women (as compared to men) obtain more total points and have more equal outcome distributions when they are walking? It is important that you provide a detailed theoretical rationale that strengthens the current manuscript. The literature support for the hypothesis about liking and mutual trust is also not substantial enough.

2. Concerns about sample size: As for now, it is unclear as to whether the current study has sufficient power to reliably study, for instance, moderation involving participants’ gender.

3. Please explain why you did not predict moderation by gender for Hypotheses 4a and 4b. I believe that strengthening the theoretical aspects of gender in negotiation will help put your work in a better shape.

4. I suggest that you should consider either HLM or SEM (preferably in the context of the Actor-Partner Interaction Model; Kenny, Kashy, & Cook, 2006) to analyse the data related to mutual liking, emotion, and mutual trust. This would allow both parties' data to be included in a single model that takes dependence (due to dyad) into account.

The reviewer reports are attached.

Sincerely,

Ricky Wong, PhD

Academic Editor, Plos ONE

Reviewer Comments to Author:

Reviewer: 1

Comments to the Author

1. Is the manuscript technically sound, and do the data support the conclusions?

The manuscript must describe a technically sound piece of scientific research with data that supports the conclusions. Experiments must have been conducted rigorously, with appropriate controls, replication, and sample sizes. The conclusions must be drawn appropriately based on the data presented. Partly

2. Has the statistical analysis been performed appropriately and rigorously? Yes

3. Have the authors made all data underlying the findings in their manuscript fully available?

The PLOS Data policy requires authors to make all data underlying the findings described in their manuscript fully available without restriction, with rare exception (please refer to the Data Availability Statement in the manuscript PDF file). The data should be provided as part of the manuscript or its supporting information, or deposited to a public repository. For example, in addition to summary statistics, the data points behind means, medians and variance measures should be available. If there are restrictions on publicly sharing data—e.g. participant privacy or use of data from a third party—those must be specified. Yes

Is the manuscript presented in an intelligible fashion and written in standard English?

PLOS ONE does not copyedit accepted manuscripts, so the language in submitted articles must be clear, correct, and unambiguous. Any typographical or grammatical errors should be corrected at revision, so please note any specific errors here. Yes

Thank you very much for offering me the opportunity to review your paper entitled “Weaving activity into work: Walking while negotiating.” There are many things to like about your paper: First of all, I was intrigued by the general idea of the manuscript to examine the influence of walking (vs. sitting) on people’s negotiation outcomes (both objective and subjective). Moreover, I greatly appreciate the attention paid to potentially differential effects for women and men. Thus, my sense is that the general topic is very interesting to both researchers and practitioners. Moreover, I found your paper easy to read and to be clear. That being said, however, I also have a number of more critical comments. Still, I hope that they are helpful as you are onto something really interesting.

1: Although I appreciate the theoretical rationales on why walking can help to support people negotiating (e.g., creativity and memory), what I was missing perhaps the most was a detailed explanation as to why, exactly, walking would be more beneficial for women negotiators. Why, specifically, should pairs of women (as compared to men) obtain more total points and have more equal outcome distributions when they are walking? As it stands, only very short shrift is given to these questions. Going forward, providing a detailed theoretical rationale would clearly help to strengthen the current manuscript.

2: Perhaps this is a matter of preference, but I would appreciate the hypotheses to be fully and clearly articulated (p. 7).

3: Why was there no moderation by gender expected for Hypotheses 4a and 4b? This idea seems to come out of nowhere, especially since there was a related moderation hypothesis for the preceding hypotheses. Once again, this issue highlights that more theoretical clarity regarding the hypotheses and their underlying rationales is needed (see my point no. 1).

4: In contrast, some information (e.g., on the sample) appeared twice. This provides an opportunity to shorten the paper to make room for greater elaboration on the theorizing.

5: Although I understand that recruiting many participants to take part in a real, interactive study (including walking) can be challenging, I am especially concerned about the current sample size. We know from past research that gender differences in negotiation typically are small or medium-sized (e.g., Kugler et al.’s meta-analysis on negotiation initiation from 2018). To detect small or medium effect sizes typically requires greater samples, especially when interactions are to be examined (see https://approachingblog.wordpress.com/2018/01/24/powering-your-interaction-2/). Thus, as there does not seem to be an a-priori power analysis, I would appreciate an inclusion of a power sensitivity analysis (see Giner-Sorolla et al.’s paper called “Power to detect what?”). As it stands, it is unclear to me whether the current study actually has sufficient power to reliably study, for instance, moderation involving participants’ gender.

6: Please provide complete tests statistics for the main regression and mixed model analyses, as well as for the follow-up tests that are meant to examine the differential effects of walking vs. sitting for women and men. As it stands, it is unclear to me whether the follow-up tests are significant as well, which would be particularly relevant information vis-à-vis the hypotheses.

Finally, thanks again for allowing me to review your paper. In my view, you are addressing a fascinating topic, so that I hope that my comments are not only critical but also helpful.

Reviewer: 2

1. Is the manuscript technically sound, and do the data support the conclusions?

The manuscript must describe a technically sound piece of scientific research with data that supports the conclusions. Experiments must have been conducted rigorously, with appropriate controls, replication, and sample sizes. The conclusions must be drawn appropriately based on the data presented. No

2. Has the statistical analysis been performed appropriately and rigorously? Yes

3. Have the authors made all data underlying the findings in their manuscript fully available?

The PLOS Data policy requires authors to make all data underlying the findings described in their manuscript fully available without restriction, with rare exception (please refer to the Data Availability Statement in the manuscript PDF file). The data should be provided as part of the manuscript or its supporting information, or deposited to a public repository. For example, in addition to summary statistics, the data points behind means, medians and variance measures should be available. If there are restrictions on publicly sharing data—e.g. participant privacy or use of data from a third party—those must be specified. No

4. Is the manuscript presented in an intelligible fashion and written in standard English?

PLOS ONE does not copyedit accepted manuscripts, so the language in submitted articles must be clear, correct, and unambiguous. Any typographical or grammatical errors should be corrected at revision, so please note any specific errors here. No

The manuscript “Weaving activity into work: Walking while negotiating” attempts to examines an interesting and promising topic. While I do believe that the manuscript and associated data can be used to make a substantive contribution, in my review, I noted several concerns and limitations, which I outline below:

1. Literature demonstrates that walking can enhance creativity and mood. This has been highlighted in both abstract and the theory part. However, why didn’t the authors measure creativity or moods as the potential mediator?

2. There is a lack of theory for studying gender as the moderator. The authors should provide substantial theoretical background for why gender can moderate the effect. As the abstract mentions, negotiation can exacerbate power differentials. However, what is the relationship between walking and power? The authors should provide more gender-related literature review that focuses on the effect of social conditioning to substantiate the theory part.

3. There is a lack of motivation for studying gender as the moderator. From abstract, introduction, and discussion sections, it is assumed that the authors want to solve the problem of long sitting at work. Then, why did the authors use the negotiation context? Negotiation context cannot cover all the other long sitting at work. The focus was then totally lost when abruptly mentioning gender. The authors should first identify the identities of the audience first. The gender scholars or the occupational health scholars? One paper better has one consistent story line. The authors should better incorporate walking as a contextualized condition for gender research.

4. There is a misalignment between the goal of the research and the methodology. “The present study’s goal was to investigate the potential for walking to improve performance on a type of work interaction.” This is too broad. The authors should narrow down their practical and theoretical focus and the literature review to only negotiation and gender. Or the authors can broaden their data to other work settings and other work performance variables. This also applies to the tones of abstract, introduction, discussion, and other overall statements that cover the purpose of the research.

5. If the research focuses on gender as the moderator, the authors must justify why they only used same-gender pairs, instead of mixed-gender pairs.

6. The hypotheses should be stated in an academic and professional way. The authors can learn from the published papers.

7. There is no literature or theoretical support for Hypothesis 2 about outcome equity. The literature support for the hypothesis about liking and mutual trust is also not substantial enough.

8. The authors should justify why they deleted the data of the participants who know their counterpart personally. In the real world, people usually know each other in the working setting.

9. The authors should describe the condition of the room where the participants sat and the details of the walking environment. For example, did the room have a table between the recruiter and the candidate to create more power differentiation (e.g., Curhan et al., 2008)? Was the walking loop in a garden or a hallway? And more importantly, the authors must validate that all the other conditions were equivalent across the experimental conditions. For example, did the participants sit in the same room and walk in the same room? If not, then what other details were different? This is because all the other conditions might influence the negotiation process, not only the walking per se. For example, the authors must rule out the possibility that the main effect came from the green plants or the sunshine, or others’ presence in the walking loop, instead of walking VS. sitting per se.

10. “New Recruit” task has another merit that the authors could have elaborated: it has power differentiation between the recruiter and the candidate. If the power is equal, the moderating effect of gender might not be salient. This brings about another issue: the authors should examine and report the individual outcome and controlled for the role in their statistical analyses. Also, the authors should further elaborate the task and process in the text, especially the integrative and distributive issues, rather than putting everything in the supplementary materials.

11. The paragraph under Sample Demographics in Results is repetitive of the second paragraph under Sample in Methods.

12. From the results and discussion, the effects of walking are not on “work performance” (as the authors claimed) or negotiation performance more specifically, but on the relational outcomes, such as emotions, mutual liking and trust. Thus, the authors might want to change the focus of the research to be relationship building in the negotiation context, instead of emphasizing “work performance.”

13. “One explanation could be that walking created a more cooperative milieu, as walking together is a synchronous activity. While in male-female pairs women cooperate more, in same gender pairs, women have been shown to cooperate less than male-male pairs (d=.16)41. Walking together may have mitigated this difference.” The present research did not use mixed-gender pairs, so it is questionable that authors claim walking mitigated “this difference.”

14. Although the authors mentioned about power distribution in other sessions for many times, the theoretical implications did not discuss the power differentiation perspective. The authors should elaborate more on this interpretation.

15. The increase of point distribution among men participants in the walking condition seems to be contradictory to the main interpretation of the other results. Does walking increase or decrease power distribution? The authors should provide sufficient theoretical interpretation to explain the seemingly contradictory results.

Reviewers' comments:

Reviewer's Responses to Questions

**Comments to the Author**

1. Is the manuscript technically sound, and do the data support the conclusions?

Reviewer #1: Partly

Reviewer #2: No

2. Has the statistical analysis been performed appropriately and rigorously? 

Reviewer #1: Yes

Reviewer #2: Yes

3. Have the authors made all data underlying the findings in their manuscript fully available?

Reviewer #1: Yes

Reviewer #2: No

4. Is the manuscript presented in an intelligible fashion and written in standard English?

Reviewer #1: Yes

Reviewer #2: No

5. Review Comments to the Author

Reviewer #1: Thank you very much for offering me the opportunity to review your paper entitled “Weaving activity into work: Walking while negotiating.” There are many things to like about your paper: First of all, I was intrigued by the general idea of the manuscript to examine the influence of walking (vs. sitting) on people’s negotiation outcomes (both objective and subjective). Moreover, I greatly appreciate the attention paid to potentially differential effects for women and men. Thus, my sense is that the general topic is very interesting to both researchers and practitioners. Moreover, I found your paper easy to read and to be clear. That being said, however, I also have a number of more critical comments. Still, I hope that they are helpful as you are onto something really interesting.

1: Although I appreciate the theoretical rationales on why walking can help to support people negotiating (e.g., creativity and memory), what I was missing perhaps the most was a detailed explanation as to why, exactly, walking would be more beneficial for women negotiators. Why, specifically, should pairs of women (as compared to men) obtain more total points and have more equal outcome distributions when they are walking? As it stands, only very short shrift is given to these questions. Going forward, providing a detailed theoretical rationale would clearly help to strengthen the current manuscript.

2: Perhaps this is a matter of preference, but I would appreciate the hypotheses to be fully and clearly articulated (p. 7).

3: Why was there no moderation by gender expected for Hypotheses 4a and 4b? This idea seems to come out of nowhere, especially since there was a related moderation hypothesis for the preceding hypotheses. Once again, this issue highlights that more theoretical clarity regarding the hypotheses and their underlying rationales is needed (see my point no. 1).

4: In contrast, some information (e.g., on the sample) appeared twice. This provides an opportunity to shorten the paper to make room for greater elaboration on the theorizing.

5: Although I understand that recruiting many participants to take part in a real, interactive study (including walking) can be challenging, I am especially concerned about the current sample size. We know from past research that gender differences in negotiation typically are small or medium-sized (e.g., Kugler et al.’s meta-analysis on negotiation initiation from 2018). To detect small or medium effect sizes typically requires greater samples, especially when interactions are to be examined (see https://approachingblog.wordpress.com/2018/01/24/powering-your-interaction-2/). Thus, as there does not seem to be an a-priori power analysis, I would appreciate an inclusion of a power sensitivity analysis (see Giner-Sorolla et al.’s paper called “Power to detect what?”). As it stands, it is unclear to me whether the current study actually has sufficient power to reliably study, for instance, moderation involving participants’ gender.

6: Please provide complete tests statistics for the main regression and mixed model analyses, as well as for the follow-up tests that are meant to examine the differential effects of walking vs. sitting for women and men. As it stands, it is unclear to me whether the follow-up tests are significant as well, which would be particularly relevant information vis-à-vis the hypotheses.

Finally, thanks again for allowing me to review your paper. In my view, you are addressing a fascinating topic, so that I hope that my comments are not only critical but also helpful.

Reviewer #2: The manuscript “Weaving activity into work: Walking while negotiating” attempts to examines an interesting and promising topic. While I do believe that the manuscript and associated data can be used to make a substantive contribution, in my review, I noted several concerns and limitations, which I outline below:

1. Literature demonstrates that walking can enhance creativity and mood. This has been highlighted in both abstract and the theory part. However, why didn’t the authors measure creativity or moods as the potential mediator?

2. There is a lack of theory for studying gender as the moderator. The authors should provide substantial theoretical background for why gender can moderate the effect. As the abstract mentions, negotiation can exacerbate power differentials. However, what is the relationship between walking and power? The authors should provide more gender-related literature review that focuses on the effect of social conditioning to substantiate the theory part.

3. There is a lack of motivation for studying gender as the moderator. From abstract, introduction, and discussion sections, it is assumed that the authors want to solve the problem of long sitting at work. Then, why did the authors use the negotiation context? Negotiation context cannot cover all the other long sitting at work. The focus was then totally lost when abruptly mentioning gender. The authors should first identify the identities of the audience first. The gender scholars or the occupational health scholars? One paper better has one consistent story line. The authors should better incorporate walking as a contextualized condition for gender research.

4. There is a misalignment between the goal of the research and the methodology. “The present study’s goal was to investigate the potential for walking to improve performance on a type of work interaction.” This is too broad. The authors should narrow down their practical and theoretical focus and the literature review to only negotiation and gender. Or the authors can broaden their data to other work settings and other work performance variables. This also applies to the tones of abstract, introduction, discussion, and other overall statements that cover the purpose of the research.

5. If the research focuses on gender as the moderator, the authors must justify why they only used same-gender pairs, instead of mixed-gender pairs.

6. The hypotheses should be stated in an academic and professional way. The authors can learn from the published papers.

7. There is no literature or theoretical support for Hypothesis 2 about outcome equity. The literature support for the hypothesis about liking and mutual trust is also not substantial enough.

8. The authors should justify why they deleted the data of the participants who know their counterpart personally. In the real world, people usually know each other in the working setting.

9. The authors should describe the condition of the room where the participants sat and the details of the walking environment. For example, did the room have a table between the recruiter and the candidate to create more power differentiation (e.g., Curhan et al., 2008)? Was the walking loop in a garden or a hallway? And more importantly, the authors must validate that all the other conditions were equivalent across the experimental conditions. For example, did the participants sit in the same room and walk in the same room? If not, then what other details were different? This is because all the other conditions might influence the negotiation process, not only the walking per se. For example, the authors must rule out the possibility that the main effect came from the green plants or the sunshine, or others’ presence in the walking loop, instead of walking VS. sitting per se.

10. “New Recruit” task has another merit that the authors could have elaborated: it has power differentiation between the recruiter and the candidate. If the power is equal, the moderating effect of gender might not be salient. This brings about another issue: the authors should examine and report the individual outcome and controlled for the role in their statistical analyses. Also, the authors should further elaborate the task and process in the text, especially the integrative and distributive issues, rather than putting everything in the supplementary materials.

11. The paragraph under Sample Demographics in Results is repetitive of the second paragraph under Sample in Methods.

12. From the results and discussion, the effects of walking are not on “work performance” (as the authors claimed) or negotiation performance more specifically, but on the relational outcomes, such as emotions, mutual liking and trust. Thus, the authors might want to change the focus of the research to be relationship building in the negotiation context, instead of emphasizing “work performance.”

13. “One explanation could be that walking created a more cooperative milieu, as walking together is a synchronous activity. While in male-female pairs women cooperate more, in same gender pairs, women have been shown to cooperate less than male-male pairs (d=.16)41. Walking together may have mitigated this difference.” The present research did not use mixed-gender pairs, so it is questionable that authors claim walking mitigated “this difference.”

14. Although the authors mentioned about power distribution in other sessions for many times, the theoretical implications did not discuss the power differentiation perspective. The authors should elaborate more on this interpretation.

15. The increase of point distribution among men participants in the walking condition seems to be contradictory to the main interpretation of the other results. Does walking increase or decrease power distribution? The authors should provide sufficient theoretical interpretation to explain the seemingly contradictory results.

6. PLOS authors have the option to publish the peer review history of their article (what does this mean?). If published, this will include your full peer review and any attached files.

Reviewer #1: No

Reviewer #2: No

---

## [Author Response · Author response to Decision Letter 0]

6 Feb 2023

Response to Reviewers

We thank the editor and both reviewers for their thorough, helpful, and constructive review of our manuscript. We have addressed all of the suggestions, substantially reframed the paper, and believe the work is much stronger as a result of this peer review process. Please find our specific responses outlined below. Thank you again for the opportunity to learn from you and better our work. 

General: 

G1: Please include your full ethics statement in the ‘Methods’ section of your manuscript file. In your statement, please include the full name of the IRB or ethics committee who approved or waived your study, as well as whether or not you obtained informed written or verbal consent. If consent was waived for your study, please include this information in your statement as well. 

G1 Response: We have now added an ethics statement to the Methods section of our paper. The Stanford University Institutional Review Board approved this study, and all participants signed a consent form prior to participation.

Editor:

E_1. Both reviewers and I have identified that a detailed explanation as to why, exactly, walking would be more beneficial for women negotiators is missing. As pointed out by R1, why, specifically, should pairs of women (as compared to men) obtain more total points and have more equal outcome distributions when they are walking? It is important that you provide a detailed theoretical rationale that strengthens the current manuscript. The literature support for the hypothesis about liking and mutual trust is also not substantial enough.

E_1 Response: We thank the reviewers for this clear and unanimous suggestion. We have substantially reframed the paper and more clearly supported the gender hypotheses. We have added a concept map of our hypotheses to show how we hypothesize walking outside together may affect the outcomes. We recognize that the literature supporting why walking together outside would improve mutual liking and trust (our measure of rapport) is limited to the references we have provided (to our knowledge). We hope this work contributes to this space, and we thought it important to investigate relational measures such as mutual liking and mutual trust as they would be helpful for future research.

E_2. Concerns about sample size: As for now, it is unclear as to whether the current study has sufficient power to reliably study, for instance, moderation involving participants’ gender.

E_2 Response: Thank you for your suggestion. We now include the paragraph below in the limitations to clarify that statistical power is a concern and the results we find should be considered exploratory. We did not do an a-priori power analysis for the proof-of-concept study. We took a reviewer’s suggestion to do a power sensitivity analysis, which we now include in the limitation sections of the paper. One worry for being under-powered is the chance that an effect was there but we did not have enough of a sample size to detect it. We did not find an effect on total points, therefore we used this as the basis for our sensitivity analysis. Based on the variances in total points we observed, a study of this size (~ 40 pairs in each condition) would be 80% powered to detect a difference by condition of approximately 1230 points (this is the minimum detectable effect). 

“The study had several limitations. First, this was a proof-of-concept study exploring the potential impact of a novel context variable (i.e. a walking meeting) on outcomes of interest. We did not do an a-priori statistical power analysis and we encourage readers to interpret our results as exploratory research. All findings should be taken as preliminary and offering inspiration for further, focused studies. Statistical power is a serious limitation for studies in this context due to high variances and low likely effect sizes45. As a power sensitivity analysis, based on the variances in total points we observed, for example, a study of this size (~ 40 pairs in each condition) would be powered to detect a difference by condition of approximately 1230 points (this is the minimum detectable effect). In particular, our preliminary finding that the impacts of walking on negotiation outcomes may differ by gender represents an interesting possible focal point for follow-up. If the effects of walking are indeed opposite for men and women, then statistical power may be better in single-gender studies than in studies (like this one) which consider both genders and try to estimate an overall effect.”

Just for contrast, (not for the text), we also calculated how many participants we would have needed to recruit if the effect size we did find in the total points were big enough for us to consider meaningful. To achieve 80% power for the primary condition comparison of sitting vs walking on total pair points in the negotiation, based on the current effect sizes in total points we found, we would need to run 892 pairs. To achieve 80% for the primary condition comparison of men x women on total pair points in the negotiation, based on the current effect sizes, we would need to run 470 pairs. Finally, for the interaction of condition x gender, the current study found an F statistic of .003. This would require an even larger sample to detect this level of an effect. 

E_3. Please explain why you did not predict moderation by gender for Hypotheses 4a and 4b. I believe that strengthening the theoretical aspects of gender in negotiation will help put your work in a better shape.

E_3 Response: Thank you for raising this concern. With the increased theoretical clarity and framing suggesting possible ways in which walking outside together can affect our outcomes by removing stereotypically threatening cues and contexts, we have now included the gender interaction in all of the hypotheses.

E_4. I suggest that you should consider either HLM or SEM (preferably in the context of the Actor-Partner Interaction Model; Kenny, Kashy, & Cook, 2006) to analyse the data related to mutual liking, emotion, and mutual trust. This would allow both parties' data to be included in a single model that takes dependence (due to dyad) into account.

E_4 Response: Thank you for the great reference. We now cite Kenny, Kashy, and Cook as our inspiration for using the linear mixed model approach of hierarchical linear modeling for the individual variables of emotion. We explain why we included pair as a random intercept for emotion. We also explain further why for the pair-level variable (mutual liking, mutual trust) analyses we did not use pair as a random intercept. Instead, we chose to look at the pair as a unit rather than two individuals within a pair, and took the average of liking and trust to reflect mutual liking and mutual trust of the pair. 

SEM would be a great idea! We chose to not use that analyses here because of concerns our proof-of-concept design was not robust enough to handle the pair-wise correlations involved. Now that we have detected signal in some of the outcomes but not others, a fully-powered hypothesis-confirmatory trial can be run to test for replication and mechanism. 

Reviewer 1:

R1: Thank you very much for offering me the opportunity to review your paper entitled “Weaving activity into work: Walking while negotiating.” There are many things to like about your paper: First of all, I was intrigued by the general idea of the manuscript to examine the influence of walking (vs. sitting) on people’s negotiation outcomes (both objective and subjective). Moreover, I greatly appreciate the attention paid to potentially differential effects for women and men. Thus, my sense is that the general topic is very interesting to both researchers and practitioners. Moreover, I found your paper easy to read and to be clear. That being said, however, I also have a number of more critical comments. Still, I hope that they are helpful as you are onto something really interesting.

R1 Response: Thank you so much for your positive words and interest, as well as your very helpful edits and comments. We have substantially revised the manuscript with your review, and hope you find the new version responsive and improved.

R1_1: Although I appreciate the theoretical rationales on why walking can help to support people negotiating (e.g., creativity and memory), what I was missing perhaps the most was a detailed explanation as to why, exactly, walking would be more beneficial for women negotiators. Why, specifically, should pairs of women (as compared to men) obtain more total points and have more equal outcome distributions when they are walking? As it stands, only very short shrift is given to these questions. Going forward, providing a detailed theoretical rationale would clearly help to strengthen the current manuscript.

R1_1 Response: We have substantially reframed the paper to be clearer about why and how walking together may improve objective and subjective negotiation outcomes for everyone. We also have added more detail explaining both why and how this may particularly help women, leading with a section describing the gender gap in negotiations sensitive to contextual shifts (such as removing cues to power differentials or having multiple issues rather than a single outcome). We hypothesize that walking together outside will benefit women more because they start out disadvantaged stereotypically compared to men. 

We also have created and included a hypotheses concept map (now Figure 1) to help provide an overview of how we think walking together outside will affect the outcomes. 

R1_2: Perhaps this is a matter of preference, but I would appreciate the hypotheses to be fully and clearly articulated (p. 7).

R1_2 Response: Thank you, we have edited the hypotheses to be more clearly articulated. As the hypotheses all shared the same stem prediction, to avoid repetition, we wrote the common stem prediction (i.e. “compared to negotiating sitting together inside, negotiating when walking together when outside would lead to improvement in:”), and then numbered the hypotheses after each outcome. We then described the moderation hypotheses at the end: “For each outcome, this effect of walking compared to sitting will be greater for women pairs than men pairs.” 

We can also articulate each hypothesis separately for each measure, repeating the stem prediction, if this would be even more helpful, and not too redundant. Example below:

H1: We hypothesized that compared to negotiating sitting together inside, negotiating when walking together when outside would lead to improvement in the objective negotiation outcome of total points, and that this improvement would be greater for women pairs than men pairs.

H2: We hypothesized that compared to negotiating sitting together inside, negotiating when walking together when outside would lead to improvement in the objective negotiation outcome of outcome equity, and that this improvement would be greater for women pairs than men pairs.

…

R1_3: Why was there no moderation by gender expected for Hypotheses 4a and 4b? This idea seems to come out of nowhere, especially since there was a related moderation hypothesis for the preceding hypotheses. Once again, this issue highlights that more theoretical clarity regarding the hypotheses and their underlying rationales is needed (see my point no. 1).

R1_3 Response: With our new reframing, organization of the literature, and conceptual hypotheses map, we had outlined how the effect of walking together may improve rapport of mutual liking and trust. For continuity of analyses, we now include gender moderation as hypotheses on every outcome. 

R1_4: In contrast, some information (e.g., on the sample) appeared twice. This provides an opportunity to shorten the paper to make room for greater elaboration on the theorizing.

R1_4 Response: Thank you, we have removed the replicated paragraph.

R1_5: Although I understand that recruiting many participants to take part in a real, interactive study (including walking) can be challenging, I am especially concerned about the current sample size. We know from past research that gender differences in negotiation typically are small or medium-sized (e.g., Kugler et al.’s meta-analysis on negotiation initiation from 2018). To detect small or medium effect sizes typically requires greater samples, especially when interactions are to be examined (see https://approachingblog.wordpress.com/2018/01/24/powering-your-interaction-2/). Thus, as there does not seem to be an a-priori power analysis, I would appreciate an inclusion of a power sensitivity analysis (see Giner-Sorolla et al.’s paper called “Power to detect what?”). As it stands, it is unclear to me whether the current study actually has sufficient power to reliably study, for instance, moderation involving participants’ gender.

R1_5 Response: The paper by Giner-Sorolla and the post on power were both great references, we thank the reviewer for these. Thank you for suggesting the power sensitivity analysis. That we did not do an a-priori power analysis – and the risk of being underpowered- is an important limitation of our study. One worry for being under-powered is the chance that an effect was there but we did not have enough of a sample size to detect it. Therefore, in the limitation section, we address this, run a sensitivity analysis, and cite the Giner-Sorolla paper as inspiration. 

We did not find an effect on total points, therefore we used this as the basis for our sensitivity analysis. Based on the variances in total points we observed, a study of this size (~ 40 pairs in each condition) would be 80% powered to detect a difference by condition of approximately 1230 points (this is the minimum detectable effect). 

We now include this important point in the limitations of our paper as well, to clarify that statistical power is a concern and the results must be considered exploratory and preliminary – an interesting possible focal point for future studies. 

“The study had several limitations. First, this was a proof-of-concept study exploring the potential impact of a novel context variable (i.e. a walking meeting) on outcomes of interest. We did not do an a-priori statistical power analysis and we encourage readers to interpret our results as exploratory research. All findings should be taken as preliminary and offering inspiration for further, focused studies. Statistical power is a serious limitation for studies in this context due to high variances and low likely effect sizes45. As a power sensitivity analysis, based on the variances in total points we observed, for example, a study of this size (~ 40 pairs in each condition) would be powered to detect a difference by condition of approximately 1230 points (this is the minimum detectable effect). In particular, our preliminary finding that the impacts of walking on negotiation outcomes may differ by gender represents an interesting possible focal point for follow-up. If the effects of walking are indeed opposite for men and women, then statistical power may be better in single-gender studies than in studies (like this one) which consider both genders and try to estimate an overall effect.”

R1_6: Please provide complete tests statistics for the main regression and mixed model analyses, as well as for the follow-up tests that are meant to examine the differential effects of walking vs. sitting for women and men. As it stands, it is unclear to me whether the follow-up tests are significant as well, which would be particularly relevant information vis-à-vis the hypotheses.

 R1_6 Response: Thank you, we now include all of the non-significant test values as well.

R1_7: Finally, thanks again for allowing me to review your paper. In my view, you are addressing a fascinating topic, so that I hope that my comments are not only critical but also helpful.

R1_7 Response. We greatly appreciate your comments and feel the manuscript is stronger because of responding to them.

Reviewer: 2

R2: The manuscript “Weaving activity into work: Walking while negotiating” attempts to examines an interesting and promising topic. While I do believe that the manuscript and associated data can be used to make a substantive contribution, in my review, I noted several concerns and limitations, which I outline below:

R2 Response: Thank you for your review and suggestions. We hope to have addressed them and believe our paper is stronger as a result of responding to your comments and thoughts.

R2_1. Literature demonstrates that walking can enhance creativity and mood. This has been highlighted in both abstract and the theory part. However, why didn’t the authors measure creativity or moods as the potential mediator?

R2_1 Response: This is a great idea. We did not have a direct measure of creative thinking in the negotiation, but we instead hypothesized that walking would help negotiation because optimal negotiations use analogical thinking (finding commonalities across disparate ideas, rather than only entertaining a single solution). The benefit then should have been seen in total pair points. An idea for a process measure to capture the cognitive benefits of negotiation while walking would be to code the actual negotiation for number of times pairs identified common ground across disparate views. We now add this to the future directions portion of the paper.

We did measure negative and positive emotions as a proxy for mood. It is a great suggestion to test mediators; we did not run a mediational analysis for this study. A follow-up, fully-powered study to measure and test mechanism would be an optimal place to use SEM analyses, and we hope this to be done in the future.

R2_2. There is a lack of theory for studying gender as the moderator. The authors should provide substantial theoretical background for why gender can moderate the effect. As the abstract mentions, negotiation can exacerbate power differentials. However, what is the relationship between walking and power? The authors should provide more gender-related literature review that focuses on the effect of social conditioning to substantiate the theory part.

R2_2 Response: Thank you for the encouragement to organize our paper with more theoretical background and motivation. We have substantially reframed the paper and now lead with a section addressing the literature showing contextually-sensitive gender gaps in negotiation outcomes. We note this now in the paper that while we do believe walking together will improve negotiation outcomes for everyone, because we believe it will remove barriers that commonly impair women’s performance, we explore the moderation of gender. We also now include a conceptual map (Figure 1) to suggest potential connections between the condition of walking outside and our outcome measures. 

R2_3. There is a lack of motivation for studying gender as the moderator. From abstract, introduction, and discussion sections, it is assumed that the authors want to solve the problem of long sitting at work. Then, why did the authors use the negotiation context? Negotiation context cannot cover all the other long sitting at work. The focus was then totally lost when abruptly mentioning gender. The authors should first identify the identities of the audience first. The gender scholars or the occupational health scholars? One paper better has one consistent story line. The authors should better incorporate walking as a contextualized condition for gender research.

R2_3 Response: This is a great point, and in response we have substantially reframed our paper to be more in line with our theoretical points, our motivation, and our hypothesized effects of walking together outside on these outcomes. We hope that our new framing addresses your valid point and situates the work for the proper audience. This work was a uniquely interdisciplinary project and study, with study investigators from business, health and medicine, education, statistics, and psychology all united by the first author and passion for this topic to be addressed; therefore, each of us learned different important things for our disciplines. However, we agree with you that the paper should instead be framed more for the audience studying the contextual effects of negotiation and gender, and hope that our new framing situates the work properly.

R2_4. There is a misalignment between the goal of the research and the methodology. “The present study’s goal was to investigate the potential for walking to improve performance on a type of work interaction.” This is too broad. The authors should narrow down their practical and theoretical focus and the literature review to only negotiation and gender. Or the authors can broaden their data to other work settings and other work performance variables. This also applies to the tones of abstract, introduction, discussion, and other overall statements that cover the purpose of the research.

R2_4 Response: We have substantially reframed the paper, rewritten the introduction and literature review, and reframed our overall purpose. Thank you very much for your suggestion to align our study goals with our audience and study design. While we had a highly interdisciplinary team, this study should target a more specific audience and purpose. We hope the edits now address this.

R2_5. If the research focuses on gender as the moderator, the authors must justify why they only used same-gender pairs, instead of mixed-gender pairs.

R2_5 Response: This is a great point. Given resource constraints for this preliminary study, we realized that we would have to prioritize among competing goals. On the one hand, a desire for completeness would have led us to a more complex design in which we had M-M pairs, F-F pairs, and M-F pairs. On the other hand, our main research question was whether there were gender differences in the effects of walking in nature (versus sitting inside) on negotiation-related outcomes and affect. As noted by the Editor (point E-2), given available resources, there was already a concern about power, and we decided that for this initial study, it would be wisest to keep the design as simple as we could (and thus include only M-M and F-F pairs), particularly in light of evidence that women may negotiate differently with men than with women, and may prioritize being liked to different degrees (Babcock and Laschever, 2003). Our decision to maintain a focused design meant that we maximized our power to detect predicted effects, and left for the future a more complete design. We see the present study as a first step in isolating what a simple shift in context could do for negative feelings and stereotype threat for women in the negotiation context. An important next study could use mixed gender pairs vs same gender as a condition variable; here, one would also have to look at the interaction of role x gender. We now include a reference justifying why we chose same-gender pairs:

Same gender pairs were employed to distill the effects of men and women negotiation from the effects of mixed-gender pairs21

R2_6. The hypotheses should be stated in an academic and professional way. The authors can learn from the published papers.

R2_6 Response: Thank you, we have edited the hypotheses to be more clearly articulated. As the hypotheses all shared the same stem prediction, to avoid repetition, we wrote the common stem prediction (i.e. “compared to negotiating sitting together inside, negotiating when walking together when outside would lead to improvement in:”), and then numbered the hypotheses after each outcome. We then described the moderation hypotheses at the end: “For each outcome, this effect of walking compared to sitting will be greater for women pairs than men pairs.” 

We can also articulate each hypothesis separately for each measure, repeating the stem prediction, if this would be even more helpful, and not too redundant. Example below:

H1: We hypothesized that compared to negotiating sitting together inside, negotiating when walking together when outside would lead to improvement in the objective negotiation outcome of total points, and that this improvement would be greater for women pairs than men pairs.

H2: We hypothesized that compared to negotiating sitting together inside, negotiating when walking together when outside would lead to improvement in the objective negotiation outcome of outcome equity, and that this improvement would be greater for women pairs than men pairs.

…

R2_7. There is no literature or theoretical support for Hypothesis 2 about outcome equity. The literature support for the hypothesis about liking and mutual trust is also not substantial enough.

R2_7 Response: We have added to the measures section to explain outcome equity as our proxy for power differential. We have also included in the introduction and on the new Figure 1 hypothesis concept map how we believe walking outside together will influence outcome equity. We added more references on behavioral synchrony and cooperation to further theoretically support how walking together may improve rapport (which we measure by mutual liking and mutual trust), but we also acknowledge there is a paucity of work in this particular area. We hope this study adds to the work! 

R2_8. The authors should justify why they deleted the data of the participants who know their counterpart personally. In the real world, people usually know each other in the working setting.

R2_8 Response: This is a good point. We include now a statement as to why we removed those who knew each other previously, as we didn’t randomize based on this and we wanted to isolate the effects of walking together on relational outcomes independent of knowing each other. 

“excluded due to… knowing their counterpart personally. The latter was a reason for exclusion because we did not a priori assess or randomize participants based on knowing each other, which would influence the relational outcomes of mutual liking and mutual trust. We also wanted to isolate the effects of walking independent of prior relationship (and did not have a pre-test measure to assess change).”

R2_9. The authors should describe the condition of the room where the participants sat and the details of the walking environment. For example, did the room have a table between the recruiter and the candidate to create more power differentiation (e.g., Curhan et al., 2008)? Was the walking loop in a garden or a hallway? And more importantly, the authors must validate that all the other conditions were equivalent across the experimental conditions. For example, did the participants sit in the same room and walk in the same room? If not, then what other details were different? This is because all the other conditions might influence the negotiation process, not only the walking per se. For example, the authors must rule out the possibility that the main effect came from the green plants or the sunshine, or others’ presence in the walking loop, instead of walking VS. sitting per se.

R2_9 Response: This is a great point, we have added detail to the contexts in the methods section. 

“Sitting condition participants sat across from each other, face-to-face in a room, with a table in between them. All sitting conditions were run in the same room. Walking condition participants were given a map of a 15-minute walking loop outside to follow. The loop was on the university campus which is a mix of trees, university buildings, alternate walking paths, and people. Variations in busy-ness, sunshine, and temperature occurred in the outdoor walking condition.”

We did not isolate the effect of outdoors (green environments and sunshine) from walking together, nor did we isolate the effect of walking from walking together. We have also added a section noting these limitations in the discussion section.

“We also did not isolate the effects of being outdoors on the current outcomes. An ideal 2x2 future study could vary outdoors (inside board room vs outside board room) by movement (sitting vs walking). While there is substantial literature supporting the mood enhancing effects of being outdoors, many of the cognitive effects of movement relevant to the current negotiation exercise is movement whether one is indoors or outdoors.”

R2_10. “New Recruit” task has another merit that the authors could have elaborated: it has power differentiation between the recruiter and the candidate. If the power is equal, the moderating effect of gender might not be salient. This brings about another issue: the authors should examine and report the individual outcome and controlled for the role in their statistical analyses. Also, the authors should further elaborate the task and process in the text, especially the integrative and distributive issues, rather than putting everything in the supplementary materials.

This is a valid point about the uniqueness of this measure. Rather than comparing individual points of recruiter to candidate in a mixed methods analysis, we chose to look at outcome equity as a proxy for the difference between the points earned. Individual points are necessarily tied to the partner’s points, therefore evaluating one person’s higher points and the other person’s necessarily lower points would in a way be double counting the effect for the pair. Therefore, we chose to use pair level outcomes for the negotiation total points and point differential. Here is the analysis on individual points however. Using a linear mixed effects model to compare the individual points for those who walked vs those who sat, we found no main effect of condition (p=.71), gender (p=.65), nor did we find a significant interaction of condition by gender (p=.97).

The idea of running recruiter x candidate analysis is excellent – we did run it, but also know we do not have enough individuals in each cell to reliably run the full factorial. Additionally, the three-way interaction is difficult to interpret. We have included the graph in the word document, formatted response to reviewers. Walking appeared to slightly decrease men recruiters, but boost men candidates. Walking appeared to boost women recruiters, but slightly decrease women candidates. This interaction is hard to derive from the literature, and with already concerns of being underpowered, we chose not to include it. One possibility is male recruiters typically hold the power positions when seated face to face, and when the dynamic was changed to walking they worked more cooperatively. Women recruiters, on the other hand, may typically fear backlash from asserting positions of power in the stereotypical context; therefore walking outside may have led them to lean into the position more.

We have added more detail about the measure into the measure section. 

“Parties must agree on eight issues: two “fixed-sum” issues where parties’ pay out are mutually opposing (one party’s gain comes at the other’s loss); two congruent issues, or “value sharing” issues where parties’ interests are aligned (both parties want the same thing); and four integrative, or “value-creating” issues where parties can make trade-offs to create value and earn more points through pairing issues with asymmetric payouts.”

R2_11. The paragraph under Sample Demographics in Results is repetitive of the second paragraph under Sample in Methods.

R2_11 Response: We have removed the duplication, thank you for noting this.

R2_12. From the results and discussion, the effects of walking are not on “work performance” (as the authors claimed) or negotiation performance more specifically, but on the relational outcomes, such as emotions, mutual liking and trust. Thus, the authors might want to change the focus of the research to be relationship building in the negotiation context, instead of emphasizing “work performance.”

R2_12 Response: This is a great point, we did not find an effect on total points, which was a feature that indicates overall performance on the task. We removed statement of this being effective on work performance (which was also too general a term to be accurate given our measure was only on negotiation). We do believe that outcome equity is an important performance measure of negotiation, as it results in the joint gains of both parties within the negotiation. We appreciate that the main findings are overall subjective – and relational, as you suggest. We added a sentence to address this observation in the discussion. 

“These findings show more of a relational impact of condition than a work performance impact.”

R2_13. “One explanation could be that walking created a more cooperative milieu, as walking together is a synchronous activity. While in male-female pairs women cooperate more, in same gender pairs, women have been shown to cooperate less than male-male pairs (d=.16)41. Walking together may have mitigated this difference.” The present research did not use mixed-gender pairs, so it is questionable that authors claim walking mitigated “this difference.”

R2_13 Response: We removed this reference to the mixed gender pairs. We also add more of the literature on how walking together induces synchrony and cooperation in the introduction.

R2_14. Although the authors mentioned about power distribution in other sessions for many times, the theoretical implications did not discuss the power differentiation perspective. The authors should elaborate more on this interpretation.

R2_14 Response. We now have connected the theoretical way in which walking together outside could reduce stereotype threat with the outcome measure of outcome equity (our measure of power differential, as a smaller difference between the two parties indicates more cooperation and mutual win, rather than single party winning the lion share of the points). We hope both the introduction and the addition of Figure 1 makes this clearer. 

R2_15. The increase of point distribution among men participants in the walking condition seems to be contradictory to the main interpretation of the other results. Does walking increase or decrease power distribution? The authors should provide sufficient theoretical interpretation to explain the seemingly contradictory results.

R2_15 Response: This was puzzling and not predicted by our hypothesis or purported theoretical mechanisms or review of the literature. We now include a candidate explanation in the discussion, but agree there is no theoretical basis for this contradiction in results.

“That walking decreased the outcome equity in men, a higher level of power differences in the outcomes, was surprising. While exercise may increase testosterone in men more than women, theoretically priming more competitive behaviors41,42, walking is not at the level of intensity as aerobic exercise. One possibility is that walking together for women motivates a cooperative mindset but walking together for men feels more novel, or may create greater vigilance and more competitiveness. Future studies should not only include measures of perceived power and perceived competitiveness, but also include a measure of walking speed, which may be faster in more competitive negotiations.”

---

## [Editor Report · Decision Letter 1]

17 Feb 2023

PONE-D-22-23834R1Moving Outside the Board Room: A Proof-of-Concept Study on the Impact of Walking while NegotiatingPLOS ONE

Dear Dr. Oppezzo,

Thank you for submitting your manuscript to PLOS ONE. After careful consideration, we feel that it has merit but does not fully meet PLOS ONE’s publication criteria as it currently stands. Therefore, we invite you to submit a revised version of the manuscript that addresses the points raised during the review process.

We look forward to receiving your revised manuscript.

Kind regards,

Ricky Siu Wong, PhD

Academic Editor

PLOS ONE

Journal Requirements:

Additional Editor Comments:

We feel that it has merit and we feel that you have taken the comments given by the reviewers and myself very seriously. Well done. However, this is a very minor issue relating to how you cite references in the main text. Please use [1]. [2], etc instead of superscripts. Once you have rectified this, I am happy to make a decision without sending out your paper for review again.

---

## [Author Response · Author response to Decision Letter 1]

17 Feb 2023

I have changed the citations to all be [1] formatted.

Thank you so much for you consideration and I am pleased that you and reviewers like the new version (so much stronger thanks to your helpful comments!)

---

## [Editor Report · Decision Letter 2]

21 Feb 2023

Moving Outside the Board Room: A Proof-of-Concept Study on the Impact of Walking while Negotiating

PONE-D-22-23834R2

Dear Dr. Oppezzo,

We’re pleased to inform you that your manuscript has been judged scientifically suitable for publication and will be formally accepted for publication once it meets all outstanding technical requirements.

Kind regards,

Ricky Wong, PhD

Academic Editor

PLOS ONE

---

## [Editor Report · Acceptance letter]

10 Mar 2023

PONE-D-22-23834R2 

Moving Outside the Board Room: A Proof-of-Concept Study on the Impact of Walking while Negotiating 

Dear Dr. Oppezzo:

I'm pleased to inform you that your manuscript has been deemed suitable for publication in PLOS ONE. Congratulations! Your manuscript is now with our production department. 

Kind regards, 

on behalf of

Dr. Ricky Siu Wong 

Academic Editor

PLOS ONE